# Unified Multi-Modal Interleaved Document Representation for Information Retrieval

## Abstract

Information Retrieval (IR) methods aim to identify relevant documents in response to a given query, which have gained remarkable attention due to their successful application in various natural language tasks. However, existing approaches typically consider only the textual information within the documents, which overlooks the fact that documents can contain multiple modalities, including texts, images, and tables. Further, they often segment each long document into multiple discrete passages for embedding, preventing them from capturing the overall document context and interactions between paragraphs. We argue that these two limitations lead to suboptimal document representations for retrieval. In this work, to address them, we aim to produce more comprehensive and nuanced document representations by holistically embedding documents interleaved with different modalities. Specifically, we achieve this by leveraging the capability of recent vision-language models that enable the processing and integration of text, images, and tables into a unified format and representation. Moreover, to mitigate the information loss from segmenting documents into passages, instead of representing and retrieving passages individually, we further merge the representations of segmented passages into one single document representation, while we additionally introduce a reranking strategy to decouple and identify the relevant passage within the document if necessary. Then, through extensive experiments on diverse information retrieval scenarios considering both the textual and multimodal queries, we show that our approach substantially outperforms relevant baselines, thanks to the consideration of the multimodal information interleaved within the documents in a unified way.

## 1 Introduction

Information Retrieval (IR) is the task of fetching relevant documents from a large corpus in response to an input query, which becomes a fundamental process to various real-world applications including web search engines and question-answering systems (Shah et al., 2019; Lewis et al., 2020; Guu et al., 2020). Specifically, to retrieve documents for the query, traditional approaches have focused on their textual representations, utilizing either sparse retrieval methods such as TF-IDF and BM25 (Robertson et al., 1994; Jones, 2004), which rely on exact term matching between the query and document, or dense retrieval methods such as DPR and ANCE (Karpukhin et al., 2020; Xiong et al., 2021), which leverage neural embeddings of the query and document text to capture semantic similarities between them over a continuous vector space. Recently, dense retrieval methods have gained more popularity over sparse methods due to their capability to capture semantic nuances and context beyond simple keyword matching, leading to multiple successes with improved performance.

Despite their huge successes, existing (dense) retrieval methods face a couple of severe challenges. First, they primarily rely on the textual data for document embedding and retrieval, overlooking the fact that modern documents often contain multimodal content, such as images and tables (beyond the plain text), which can carry critical information that may be essential for accurately understanding and retrieving the relevant documents. To be specific, a diagram within a medical article can more effectively represent the structure of a molecule or the progression of a disease, offering more clarity that would be difficult to achieve with text alone, and omitting such multimodal content can lead to an incomplete understanding (and potentially inaccurate retrieval) of the documents. Also, the segmentation of long documents into discrete passages, which is commonly employed by retrieval models to handle the length limitation for embeddings, may prevent models from capturing the full

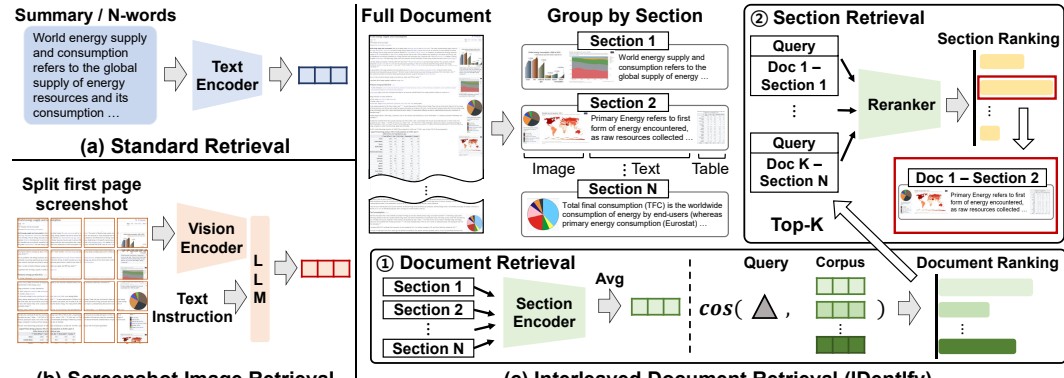

Figure 1: Comparison of different IR approaches. **(a)**: Conventional methods use a small portion of the text within the document for its representation. **(b)**: Recent methods use first-page screenshot images to represent the document. **(c)**: Our approach leverages the full contextual information within documents interleaved with multiple modalities by considering them in their original format, and is capable of pinpointing relevant sections.

context and the intricate relationships between different parts of the document, ultimately leading to suboptimal retrieval performance. It is worthwhile noting that, concurrent to our work, while there has been recent work that screen captures the document and then embed its screenshots (to consider different modalities in a unified format) (Faysse et al., 2024; Ma et al., 2024a), not only its content (such as paragraphs, images, and tables) can be fragmented into different sub-images, leading to the loss of contextual coherence across the entire document, but also the visual representation of text may hinder the model's ability to capture the semantic relationships present in the original textual data, while increasing the image resolution leads to the critical concern on the memory requirements.

In this work, we introduce a novel approach to holistic document embedding for IR, which addresses the aforementioned challenges by representing and retrieving the documents interleaved with different modalities in a unified manner. Specifically, our method revolves around the recent advance of Vision-Language Models (VLMs), which enables the processing and integration of multimodal content (such as text, images, and tables) directly into a single token sequence, thereby preserving the context and relationships between various parts of the document, unlike the previous approaches that rely on the fragmented visual representations. Furthermore, in cases where the number of tokens in a document is large and exceeds the capacity of a single context window of VLMs, we propose a strategy to segment the document into manageable passages, each represented within the token limit, and combine these passage representations into a unified document representation, which differs from existing IR approaches that independently represent and retrieve at the passage level, potentially losing the overall document context. Lastly, to accurately identify only the relevant sections within the lengthy documents, we introduce a reranking mechanism that is trained to pinpoint the passage most pertinent to the query (among all the other passages within the document), effectively allowing for both the coarse-grained document-level matching and the fine-grained passage-level retrieval. We provide the visual illustrations of the overall pipeline of IDentIfy against prior work in Figure 1. We refer to our overall method as **I**nterleaved **D**ocum**ent** **I**n**f**ormation Retrieval S**y**stem (IDentIfy).

We experimentally validate the effectiveness of IDentIfy on four different benchmark datasets, considering both the text-only and multimodal queries. On a battery of tests conducted, we then observe that our approach substantially outperforms relevant baselines that consider only the uni-modality for document representations, thanks to the consideration of multimodal content. Further, we find that the strategy to represent the whole document with its single representation (by merging embeddings of its splits if necessary) is superior to the approach of individually representing them for document retrieval, but also performing reranking over the sections of the retrieved document is superior to the approach of directly retrieving those sections, which confirm the efficacy of the proposed retrieval and reranking pipeline for document and passage retrieval, respectively.

## 2 RELATED WORK

**Information Retrieval** Information Retrieval (IR) is the task of accurately finding documents relevant to a given query from a large corpus, such as Wikipedia, which has been a crucial component for a variety of applications, including search engines, question-answering systems, and conversational agents (Zhu et al., 2023; Gao et al., 2023; Ram et al., 2023; Shi et al., 2024; Jeong et al., 2024a).

Specifically, to retrieve the relevant documents, earlier IR approaches measured similarity between queries and documents based on their lexical term matching, such as BM25 and TF-IDF (Robertson et al., 1994; Jones, 2004). Yet, these methods often struggled to capture the semantic nuances beyond surface-level term overlaps. To overcome this, along with advancements in language models (Devlin et al., 2019; Liu et al., 2019). there has been dense retrieval approaches that embed both the queries and documents into a shared dense vector space (Karpukhin et al., 2020; Xiong et al., 2021), enabling the calculation of semantic similarity between them more effectively by capturing the deeper contextual information. However, previous IR studies have mainly focused on enhancing the textual representations of queries and documents, while overlooking the fact that documents often consist of diverse modalities (such as images and tables) beyond text, which can potentially provide richer context and aid in more accurate retrieval (Liu et al., 2021; Jeong et al., 2024b).

**Multimodal Information Retrieval** Recent studies in IR have expanded the focus from purely text-based retrieval models to those that consider other modalities, such as images (Radford et al., 2021; Xiao et al., 2024), tables (Herzig et al., 2021; Chen et al., 2024) and graphs (Baek et al., 2023); however, the majority of these approaches (Zhou et al., 2024; Long et al., 2024; Lerner et al., 2024; Nowak et al., 2024; Caffagni et al., 2024) have primarily explored how to process the multimodal *queries*, meanwhile, they often overlook the equally important multimodal characteristics of the *documents* being retrieved. Specifically, we argue that, while incorporating multimodal elements in queries has expanded the range and diversity of query types that IR systems can handle, considering the multimodal nature of the documents can lead to a more holistic representation of retrieval targets, which can ultimately lead to enhancing the overall retrieval performance. In efforts to handle diverse multimodal elements within documents, there are concurrent studies that have proposed to capture screenshots of documents, such as PDFs (Faysse et al., 2024) or Wikipedia web pages (Ma et al., 2024a), and subsequently encoding them through vision models (Ding et al., 2024). However, these methods are not only limited by factors, such as image resolution and computational memory, constraining their application to documents longer than a single page[1], but also fall short by treating the diverse modalities within a document as a single visual entity, leading to suboptimal document representations that fail to effectively capture the nuanced interdependence between text and images. Furthermore, they do not address the critical issue of splitting documents into smaller fragments (e.g., sub-images), which may disrupt the holistic contextual view of the entire document.

**Vision-Language Models** Recent Vision-Language Models (VLMs) have emerged as a powerful tool for jointly processing visual and textual data, combining the image understanding capabilities of visual encoders (Radford et al., 2021; Zhai et al., 2023) with the advanced reasoning abilities of language models (OpenAI, 2022; 2023a). These models have achieved remarkable performance across diverse vision-language (VL) tasks (such as image captioning and visual question answering) (Dai et al., 2023; OpenAI, 2023b), with the substantially limited attention on their applications to IR. We note that the latest developments in this field have particularly focused on enabling VLMs to handle interleaved, multimodal content, which involves a mixed sequence of images and text (Zhang et al., 2023; Li et al., 2024). In particular, LLaVA-NeXT-Interleave (Li et al., 2024) introduces a fine-tuning approach that specifically enhances the VLMs' capacity to understand complex interleavings of multiple images and text within a single context. Drawing inspiration from these advances, in this work, we propose to harness the capabilities of VLMs to create unified embeddings for documents interleaved with text and images (as well as tables) for IR, which is a big shift from even the recent IR approach (Ma et al., 2024b) that still embeds the documents with the recent but text-based models like Llama (Touvron et al., 2023a;b), failing to fully capture the diverse multimodal content.

## 3 METHOD

We present IDentIfy to holistically represent documents interleaved with multimodal elements.

### 3.1 PRELIMINARY

We begin with preliminaries, formally explaining information retrieval and vision-language models.

**Information Retrieval** Recall that Information Retrieval (IR) is the task of searching for relevant documents from a large corpus in response to a given query. Formally, let $q$ denote a query, $d$ denote

---

[1] For instance, Ma et al. (2024a) requires processing 9.8k image tokens just to process a single-page document, and it results in 2TB of memory for handling the entire Wikipedia corpus, which is not much practical.

a document, and $\mathcal{D}$ denote a collection of documents ($\boldsymbol{d} \in \mathcal{D}$), where each query and document can be represented as a sequence of tokens: $\boldsymbol{q} = [q_1, q_2, \ldots, q_n]$ and $\boldsymbol{d} = [d_1, d_2, \ldots, d_m]$ where $[\cdot]$ indicates a concatenation operation in a specific order. We note that traditional IR approaches typically consider these tokens as purely textual elements; however, in this work, we propose to extend this assumption to have the tokens of both the textual and visual content, to capture the multimodal nature of many real-world documents. Then, this new extension raises important questions of how can both the textual and visual content be represented within a unified token framework, and how can these multimodal tokens be seamlessly integrated and encoded for document representations. To answer those two questions, we harness the power of recent vision-language models below.

**Vision-Language Models** We now turn to describing Vision-Language Models (VLMs), which are designed to jointly encode the textual and visual information in a unified token framework. We note that these models are generally comprised of two main components: a visual encoder and a language model, interconnected through a projection layer. Specifically, given an input document that may contain interleaved modalities (e.g., text and images), the visual encoder extracts high-level visual features from (multiple) images embedded within the document, mapping them into a latent space. Then, these visual features are transformed into a sequence of visual tokens via the projection layer, represented as follows: $\mathbf{V} \in \mathbb{R}^{V \times d_{\text{emb}}}$ where $V$ denotes the visual token length and $d_{\text{emb}}$ is the token dimension size. Similarly, for the textual content embedded within the document alongside images, the language model uses a word embedding layer to convert the input text into a sequence of text tokens, represented as follows: $\mathbf{L} \in \mathbb{R}^{L \times d_{\text{emb}}}$ where $L$ denotes the token length of text.

In this work, we also propose to account for tables that are an integral modality for holistically representing the full content of documents. However, in contrast to text and images that have dedicated processing layers within the VLM architectures, tables do not have a specific representation layer. Nevertheless, we argue that recent VLMs are pre-trained on diverse web data, and subsequently they are implicitly learned to handle the table structures formatted in HTML. Consequently, we treat HTML-format table data as a linearized sequence of HTML words, applying the same word embedding layer as is used for plain text. To be formal, this process converts the table content into table tokens, as follows: $\mathbf{T} \in \mathbb{R}^{T \times d_{\text{emb}}}$ where $T$ is the token length of the table. Lastly, once extracted, the visual tokens, text tokens, and table tokens are concatenated (to form a unified token sequence) and then passed through the remaining layers of VLMs, to capture both uni- and cross-modal relationships across different modalities, enabling the comprehensive understanding of the input document.

## 3.2 RETRIEVER

We now turn to explaining how we design a retriever specifically tailored for multimodal interleaved document retrieval. In particular, to effectively retrieve documents that contain multiple modalities, our approach leverages a VLM capable of processing text, images, and tables within a single document. Further, following the standard practice of existing retrieval architectures (Karpukhin et al., 2020; Xiong et al., 2021), we use a dual-encoder structure, which consists of a query encoder and a section encoder, both are based on the VLM, which is illustrated in Figure 2 (a).

Specifically, thanks to the use of the VLM, our query encoder can take either purely textual queries $\boldsymbol{q} = \mathbf{L}_{\text{Q}}$ or multimodal queries consisting of text and corresponding visual elements $\boldsymbol{q} = [\mathbf{V}_{\text{Q}}, \mathbf{L}_{\text{Q}}]$. Also, to obtain the final query representation, we introduce a learnable token called 'End of Query', $[\text{EoQ}] \in \mathbb{R}^{d_{\text{emb}}}$. This token is appended to the end of the sequence of query tokens $\boldsymbol{q}$, and the final concatenated tokens $[\boldsymbol{q}, [\text{EoQ}]]$ are then passed through the query encoder. Then, the model output corresponding to $[\text{EoQ}]$ is used as the final query representation, as follows: $\mathbf{Z}_{\text{Q}} \in \mathbb{R}^{d_{\text{emb}}}$.

For documents, we first represent each document $\boldsymbol{d}$ as a sequence of sections $\boldsymbol{d} = [\boldsymbol{s}_i]_{i=1}^{S}$ (with a total of $S$ sections), where each section $\boldsymbol{s}_i$ is derived by dividing the document according to the subtitles in the document. $\boldsymbol{s}_i$ can contain a combination of text tokens $\mathbf{L}_{\text{S}i}$, visual tokens from embedded images $\mathbf{V}_{\text{S}i}$, and table tokens $\mathbf{T}_{\text{S}i}$, denoted as follows: $\boldsymbol{s}_i = [\mathbf{V}_{\text{S}_i}, \mathbf{L}_{\text{S}_i}, \mathbf{T}_{\text{S}_i}]$. Then, to obtain a section-level representation, similar to the query representation, we introduce a learnable token, called 'End of Section': $[\text{EoS}] \in \mathbb{R}^{d_{\text{emb}}}$, which is similarly appended at the end of each section. We then forward the concatenated tokens $[\boldsymbol{s}_i, [\text{EoS}]]$ to the section encoder, and, after that, the output corresponding to $[\text{EoS}]$ is used to form the section representation, as follows: $\mathbf{Z}_{\text{S}_i} \in \mathbb{R}^{d_{\text{emb}}}$. Additionally, the overall document representation is obtained by averaging the representations of all sections within the document, defined as follows: $\mathbf{Z}_{\text{D}} = \frac{1}{S} \sum_{i=1}^{S} \mathbf{Z}_{\text{S}_i}$.

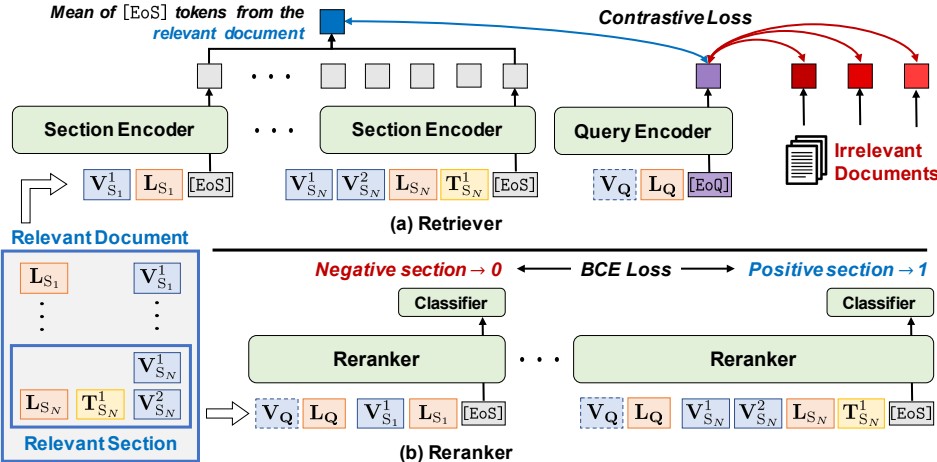

Figure 2: Overview of IDentIfy. **(a)**: In our document retriever, a query encoder represents a query (purple), and sections are encoded with a section encoder whose embeddings averaged to form a document representation (blue). Contrastive learning loss (red) is used for training the document retriever. **(b)**: Reranker scores query-section relevance with the concatenation of the query and section, trained using Binary Cross-Entropy loss.

The remaining step to discuss here is how to train those two query and document retrievers for IR. Recall that the goal of the retriever is to assess a relevance score between the query and the document. To achieve this goal, we use a contrastive learning loss based upon the query and document representations, whose objective is to assign higher similarity scores to relevant documents (positive samples) and lower scores to irrelevant ones (negative samples) for the query, formulated as follows:

$$\mathcal{L}_{\text{retriever}} = -\frac{1}{B}\sum_{i=1}^{B}\log\left(\frac{\texttt{sim}(\mathbf{Z}_{\text{Q}_i},\mathbf{Z}_{\text{D}_i})}{\sum_{j=1}^{B}\texttt{sim}(\mathbf{Z}_{\text{Q}_i},\mathbf{Z}_{\text{D}_j})}\right), \quad \texttt{sim}(\mathbf{Z}_{\text{Q}},\mathbf{Z}_{\text{D}}) = \frac{\mathbf{Z}_{\text{Q}}^{\top}\mathbf{Z}_{\text{D}}}{\|\mathbf{Z}_{\text{Q}}\|\|\mathbf{Z}_{\text{D}}\|},$$

where $B$ is the batch size during the training phase. Here, by minimizing $\mathcal{L}_{\text{retriever}}$, the retriever learns to optimize the similarity between queries and their relevant documents, enabling the retrieval of the most pertinent documents (among all) for the given input query during inference.

### 3.3 RERANKER

To enable fine-grained retrieval within documents beyond the retrieval of documents themselves, we introduce a section-level reranking mechanism that identifies the section most relevant to the input query. In particular, once the document is retrieved, the objective of the reranker $f_{\text{R}}$ is to pinpoint the specific sections within the document that best match the query. We also note that this reranker is similarly operationalized with a single VLM along with a binary classifier on top of it, which directly measures the relevance of each query-section pair, illustrated in in Figure 2 (b).

Formally, for a retrieved document, we take each of its sections $\boldsymbol{s}_i$ and concatenate it with the query $\boldsymbol{q}$ and a learnable token for section embedding $[\texttt{EoS}]$, forming the input sequence of $[\boldsymbol{q}, \boldsymbol{s}_i, [\texttt{EoS}]]$. The concatenated tokens are then processed through the reranker, and the model output corresponding to $[\texttt{EoS}]$ captures the relevant between the query and section, which is further subsequently passed to a binary classifier consisting of a linear layer followed by a Sigmoid function. Through this process, the classifier outputs a probability score indicating the likelihood of the section being relevant to the query, i.e., a score close to one denotes a high relevance (positive section), meanwhile, a score near zero indicates irrelevance (negative section).

To train this reranker, we use the Binary Cross-Entropy (BCE) loss, formalized as follows:

$$\mathcal{L}_{\text{reranker}} = \sum_{i=1}^{B}\sum_{j=1}^{S_i}\frac{1}{BS_i}\ell\left(\mathbf{y}_{(\text{S}_{i,j})}, f_{\text{R}}\left([\boldsymbol{q}, \boldsymbol{s}_{i,j}, [\texttt{EoS}]]\right)\right), \quad \ell\left(y,\hat{y}\right) = -\left[y\log\hat{y} + (1-y)\log(1-\hat{y})\right],$$

where $S_i$ is the number of sections in the $i$-th document, $\mathbf{y}_{(\text{S}_{i,j})}$ is the label for the $j$-th section of the $i$-th document $\boldsymbol{s}_{i,j}$ (with its value of one if relevant to the query $\boldsymbol{q}$ and zero otherwise), and $B$ is the batch size during training. Also, in this training process, the sections not labeled as relevant to the query are considered negative samples. Then, by minimizing $\mathcal{L}_{\text{reranker}}$, the reranker learns to predict section relevance for any query, thus refining our overall retrieval process by allowing the retrieval of not just whole documents but also their most relevant sections, for multiple use cases of IR.

## 4 EXPERIMENTS

### 4.1 DATASETS

To evaluate the effectivenss of IDentIfy, we focus on multimodal IR tasks that require understanding of both the textual and visual cues within queries and documents, which align well with our goal of enhancing retrieval of multimodal interleaved documents. The datasets considered are as follows:

**Encyclopedic-VQA** (Mensink et al., 2023) is a large-scale visual question-answering (VQA) benchmark dataset, widely used for measuring the performance of multimodal IR models. Each query is linked to a specific section of a Wikipedia document (containing an answer for it) and is manually annotated by humans. Also, this dataset offers both text-only and multimodal queries. In addition to this, the queries are related to fine-grained properties of species and landmarks. Our experiments focus on the single-hop category where questions that can be answered in a single retrieval step.

**InfoSeek** (Chen et al., 2023) is a dataset designed for knowledge-intensive VQA, covering a wide range of entities (such as landmarks, animals, and food). Questions are generated by filling human-written templates with knowledge triples (subject, relation, object) available from Wikidata, which involve only the multimodal queries. As the test dataset is not available, we use the validation set as our test set, and split the training set into training and validation subsets with a 9:1 ratio.

**ViQuAE** (Lerner et al., 2022) is a dataset focused about human entities. It provides both text-based and multimodal queries, with each query linked to a specific section of a Wikipedia document that contains an answer (annotated by humans), which makes it an idea benchmark for section retrieval.

**Open-WikiTable** (Kweon et al., 2023) is an extension of WikiSQL (Zhong et al., 2017) and WikiTableQuestions (Pasupat & Liang, 2015), designed for open-domain table question answering that requires retrieval of the most relevant table from a broader corpus. For our experiments, we adapt this dataset, aiming at identifying the document or document section containing the target table, and correspondingly, utilize the WikiTableQuestions subset of Open-WikiTable that has labels for it.

### 4.2 IMPLEMENTATION DETAILS

**Model Training and Evaluation**   We use LLaVA-NeXT-Interleave (Li et al., 2024) of 0.5B parameters as the basis VLM, for both the retriever and reranker. To take the advantage of larger batch sizes (while reducing GPU memory usage), we apply LoRA (Hu et al., 2022). Also, to further optimize the GPU usage, we combine four images into one, scaling each down to half of its original height and width. During retriever and reranker training, we consider four sections per document in representing documents and selecting negative samples. In contrast, during inference, we consider all sections within each document. For section retrieval, the top 25 documents retrieved are split into sections and passed to the rerankers. All experiments are conducted using a single H100 GPU.

**Baselines**   We compare our approach against a variety of IR baselines designed to capture different document representations. We start with Entity and Summary baselines, which are trained to retrieve documents based on their titles and summary sections. Next, we consider the Text-document retriever, which retrieves documents based on their textual content. Additionally, to consider a visual component, we consider the Single-image baseline, incorporating the first document image.

**Evaluation Metrics**   We evaluate the performance of the retriever and reranker with standard metrics: Recall@K (R@K) and Mean Reciprocal Rank@K (MRR@K). First, R@K measures whether the relevant document or section is retrieved within the top-K results. MRR@K evaluates the ranking quality by measuring the position of the first relevant item among the top-K retrieved results.

### 4.3 RESULTS AND DISCUSSION

**Interleaved format improves document retrieval.**   First, we report the retrieval performance on the Encyclopedic-VQA dataset in Table 1, where each query consists of both image and text content. From this, we observe that our approach achieves the best performance, with R@1 score improvements of 53.0%, 64.0%, and 25.0% compared to the Summary, Text-document, and Single-image retrieval baselines, respectively. The MRR@10 score similarly shows significant gains, with improvements of 36.1%, 48.5%, and 16.2% over the same baselines. This demonstrates the effectiveness of our approach in incorporating the interleaved multimodal format for document representations.

Table 1: Comparison results of the document formats for document retrieval with multimodal queries. Entity uses the document title, Summary uses its first summary section, and Text-document uses only textual content. Based on the Text-document, we add a single image (+ Single-image) or interleaved multimodal content (+ Interleaved).

| Format | R@1 | R@10 | R@100 | MRR@10 |
|---|---|---|---|---|
| Entity | 3.1 | 15.5 | 39.7 | 6.1 |
| Summary | 13.4 | 41.3 | 66.5 | 21.6 |
| Text-document | 12.5 | 37.8 | 68.7 | 19.8 |
| + Single-image | 16.4 | 45.4 | 77.1 | 25.3 |
| + Interleaved (Ours) | **20.5** | **50.0** | **78.0** | **29.4** |

Table 2: Investigation of retrieval granularity in information retrieval. Two granularities are tested on section retrieval with multimodal queries: Passage splits a document by section boundaries, and, using the split passages as retrieval targets, trains the retriever; meanwhile, Document uses documents as retrieval targets. The same reranker is applied to items retrieved by each method. * indicates a result that does not leverage reranking.

| Granularity | R@1 | R@10 | R@20 | MRR@10 |
|---|---|---|---|---|
| Passage* | 3.9 | 16.9 | 22.0 | 7.5 |
| Passage | 28.6 | 36.4 | 37.8 | 31.2 |
| Document (Ours) | **35.1** | **50.8** | **53.6** | **40.3** |

Table 3: Performance in document retrievals. **(a)**: Results of retrieval for multimodal queries on InfoSeek and ViQuAE. **(b)**: Results of retrieval for textual queries on Encyclopedic-VQA (Enc-VQA) and ViQuAE.

**(a) Document Retrieval with Multimodal Queries**

| Foramt | Dataset | R@1 | R@10 | R@100 | MRR@10 |
|---|---|---|---|---|---|
| Text-document | InfoSeek | 6.8 | 23.6 | 52.5 | 11.2 |
| + Interleaved | | **10.2** | **30.4** | **57.3** | **15.7** |
| Text-document | ViQuAE | 13.5 | 40.4 | 67.4 | 20.9 |
| + Interleaved | | **17.5** | **46.0** | **69.4** | **26.3** |

**(b) Document Retrieval with Textual Queries**

| Format | Dataset | R@1 | R@10 | R@100 | MRR@10 |
|---|---|---|---|---|---|
| Text-document | Enc-VQA | 62.7 | 76.3 | 87.4 | 67.0 |
| + Interleaved | | **65.4** | **76.8** | **87.8** | **69.0** |
| Text-document | ViQuAE | 55.8 | 71.5 | **83.0** | 60.9 |
| + Interleaved | | **56.5** | **72.2** | **83.0** | **61.6** |

To further understand the source of these performance gains, we explore two levels of retrieval granularity: passages and documents. Specifically, the passage retriever uses individual sections of documents as retrieval units, while the document retriever treats entire documents as single units. Both models are trained on the Encyclopedic-VQA dataset for multimodal retrieval. Then, we use the same reranker to both sets of results from passage and document retrievers, to directly compare their performance. In Table 2, we observe that relying solely on the passage retriever (Passage*) results in suboptimal retrieval performance, highlighting the challenge in pinpointing the most relevant section within a document using traditional retrieval methods. In contrast, when the reranker is used alongside the document retriever, the performance significantly surpasses that of the passage retrieval, achieving a 22.7% improvement in R@1 and a 29.2% improvement in MRR@10, even though the document retriever provides eight times fewer retrieval units to the reranker. These results confirm the importance of leveraging holistic context from multiple, interrelated sections within documents. In addition to this, these findings also demonstrate the notable advantages of using the interleaved multimodal elements within documents, emphasizing the potential of this direction.

**Interleaved format enhances document retrieval across modalities.** We further expand our experiments to two additional IR datasets, the InfoSeek and ViQuAE. As shown in Table 3 (a), our proposed retriever consistently surpasses the Text-document baseline in document retrieval with multimodal queries. Specifically, this leads to 50.0% and 29.6% improvements in the R@1 score, and 40.2% and 25.7% improvements in the MRR@10 score for the InfoSeek and ViQuAE, respectively. We also examine the impact of interleaved documents on textual retrieval tasks, where queries consist solely of text, and report the results in Table 3 (b). Then, the results demonstrate that the interleaved format offers advantages in retrieval of textual queries as well, resulting in 4.3% and 1.3% improvements in the R@1 score and 3.0% and 1.1% improvements in the MRR@10 score for the Encyclopedic-VQA and ViQuAE, respectively. We attribute these gains to the integration of multimodal content within documents, enabling the VLM to capture the multimodal alignment and to exploit its pre-existing knowledge for more effective document representations (Xu et al., 2024).

**Interleaved format is also beneficial in section retrieval.** Similarly, we evaluate the efficacy of our approach in section retrieval across both multimodal and textual queries, using the Encyclopedic-VQA and ViQuAE datasets. First, in section retrieval with multimodal queries, our model attains 4.2% improvement in the R@1 score and 3.3% improvement in the MRR@10 score for the Encyclopedic VQA, as shown in Table 4 (a). Similarly, in section retrieval with textual queries, our model achieves 2.3% and 7.5% improvements in the R@1 score and 1.8% and 4.9% improvements in the

Table 4: Performance of section retrievals. **(a)**: Results of retrieval for multimodal queries on Encyclopedic-VQA (Enc-VQA) and ViQuAE. **(b)**: Results of retrieval for textual queries on Enc-VQA and ViQuAE. The same document retrieval outcomes are used in section retrieval to solely measure the reranker's performance.

| (a) Section Retrieval with Multimodal Queries | | | | | |
|---|---|---|---|---|---|
| Format | Dataset | R@1 | R@10 | R@20 | MRR@10 |
| Text-document + Interleaved | Enc-VQA | 40.7 **42.4** | 52.8 **53.6** | 55.5 **55.7** | 44.8 **46.3** |
| Text-document + Interleaved | ViQuAE | **12.6** 11.4 | 31.7 **32.1** | 37.7 **39.2** | **18.2** 17.5 |

| (b) Section Retrieval with Textual Queries | | | | | |
|---|---|---|---|---|---|
| Format | Dataset | R@1 | R@10 | R@20 | MRR@10 |
| Text-document + Interleaved | Enc-VQA | 68.1 **69.7** | 79.4 **80.1** | 80.2 **80.6** | 72.3 **73.6** |
| Text-document + Interleaved | ViQuAE | 27.8 **29.9** | 50.2 **50.9** | 57.7 **59.8** | 35.0 **36.7** |

Table 5: Document and section retrieval results for tables, where Zero-shot denotes a model finetuned on Encyclopedic-VQA but not trained on the target dataset. Finetuned refers to additional training of the model on the target dataset. **(a)**: Results for tabular document retrieval on the Open-WikiTable (OWT) dataset. **(b)**: Textual section retrieval results on the ViQuAE dataset and tabular section retrieval results on the OWT dataset. **(c)**: Reranker accuracy (Acc@1) of a classification task that identifies the section containing the query-associated table within a gold document containing multiple tables. Random indicates random selection in the task.

| (a) Document Retrieval for Tables | | | | |
|---|---|---|---|---|
| Model | R@1 | R@10 | R@100 | MRR@10 |
| Zero-shot | 29.4 | 58.0 | 86.0 | 38.1 |
| Finetuned | **55.8** | **84.1** | **93.5** | **66.1** |

| (c) Tabular Classification | | | |
|---|---|---|---|
| Model | Random | Zero-shot | Finetuned |
| **Acc@1** | 11.9 | 9.3 | **56.5** |

| (b) Section Retrieval for Tables | | | | | | |
|---|---|---|---|---|---|---|
| Model | Modality | Dataset | R@1 | R@10 | R@20 | MRR@10 |
| Zero-shot Finetuned | Text | ViQuAE | 20.3 **29.9** | **49.0** 50.9 | **57.7** 59.8 | 28.9 **36.7** |
| Zero-shot Finetuned | Table | OWT | 5.9 **8.4** | **20.5** 36.7 | **29.4** 52.8 | 9.1 **15.2** |

MRR@10 score for the Encyclopedic and ViQuAE datasets, as shown in Table 4 (b). Overall, the design of our Interleaved rerankers exhibit superior or comparable performance to the Text-document rerankers. However, since the rerankers assess query relevance using a single section, they may lack access to broader contextual information from a document, which limits the potential performance gain compared to the retrievers. Nonetheless, the multimodal content interleaved within documents improves the reranker's ability to evaluate the relevance of the query to individual sections.

**Information retrieval of tabular contents in interleaved documents is challenging.** We explore a retrieval task for tabular data, whose goal is to select the document or section containing the target table relevant to the input query. Specifically, we use the Open-WikiTable dataset to train the retriever and reranker, and then compare these trained models (Finetuned) with the models trained on the Encyclopedic-VQA dataset (Zero-shot). Then, as shown in Table 5 (a), despite Open-WikiTable consisting of only 3.2k training samples, the Finetuned retriever achieves strong retrieval performance. Meanwhile, the Zero-shot retriever demonstrates only about half of the R@1 score and the MRR@10 score of the Finetuned retriever, though it remains competitive in R@100.

In contrast, the performance trends for the rerankers exhibit notable differences. The discrepancies in R@10 and R@20 scores between the Zero-shot and Finetuned retrievers, denoted in color red in Table 5 (b), are much more pronounced in the Open-WikiTable (table retrieval) experiments than the ones in the ViQuAE (text retrieval) experiments. This highlights a substantial difference between textual and tabular modalities, despite both being represented using word tokens. This suggests that these two modalities may require different handling for retrieval, which we leave as future work.

Notably, the R@1 scores for tabular section retrieval are significantly lower than those for textual section retrieval (Table 5 (b)). To better understand the difficulty of identifying the query-relevant table, we use the Open-WikiTable dataset and design a classification task. In this task, both Zero-shot and Finetuned rerankers are provided with a golden document — a document containing the target of the input query — and should identify the section that contains the target table. Notably, this setup isolates the reranker's ability to locate the target within a golden document. To ensure that the difficulty of the task is accurately assessed, we focus on documents containing multiple tables. Then, as shown in Table 5 (c), the Zero-shot reranker performs similarly to a random selection, failing to find the correct section. This accounts for its low R@10 and R@20 scores in the tabular retrieval task (Table 5 (b)). In aggregate, while the Finetuned reranker shows improved performance, it still misclassifies nearly half of the tables due to the high similarity between tables within the same

Figure 3: Trade-off between retrieval performance and training cost.

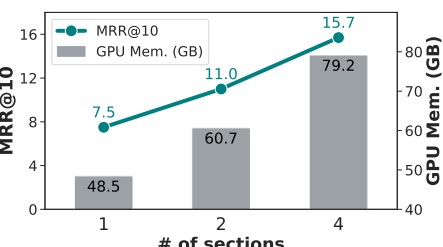

Table 6: Impact of negative sample selection in the reranker training. Top-K selects the top-k retrieved sections from the retriever as negatives. In-batch uses negatives from other sections in the same batch. In-document selects negatives from sections within the document containing the positive section.

| Negative | R@1 | R@10 | R@20 | MRR@10 |
|---|---|---|---|---|
| Top-K | 38.1 | 53.7 | 55.3 | 44.4 |
| In-batch | 39.5 | **53.8** | 55.4 | 45.0 |
| In-document (Ours) | **42.4** | 53.6 | **55.7** | **46.3** |

Table 7: Performance of different reranker designs. Contrastive follows the same training pipeline used for retrievers but here it uses sections for retrieval. Document+BCE concatenates the input query with multiple sections gathered from the same document and uses BCE loss to train the re-ranker. Section+BCE concatenates the query with each section of the document, and the re-ranker is trained with the BCE loss.

**(a) Section Retrieval for Multimodal Queries**

| Train Loss | R@1 | R@10 | R@20 | MRR@10 |
|---|---|---|---|---|
| Contrastive | 3.6 | 15.0 | 21.3 | 6.5 |
| Document+BCE | 13.6 | 29.6 | 32.9 | 24.1 |
| Section+BCE (Ours) | **42.4** | **53.6** | **55.7** | **46.3** |

**(b) Section Retrieval for Textual Queries**

| Train Loss | R@1 | R@10 | R@20 | MRR@10 |
|---|---|---|---|---|
| Contrastive | 13.6 | 37.7 | 45.1 | 20.6 |
| Document+BCE | 23.8 | 43.4 | 47.2 | 39.1 |
| Section+BCE (Ours) | **69.7** | **80.1** | **80.6** | **73.6** |

document. When combined with tables from other documents, this further complicates the task of identifying the exact query-relevant table, as shown in Finetuned reranker's low score in Table 5 (b).

## 4.4 FURTHER ANALYSIS AND ABLATION

**More sections enhance document retrieval performance but raise computational costs.** In Table 1 and Table 2, we observe that using the comprehensive multimodal content and enriched contextual information significantly improves document retrieval performance. Accordingly, we anticipate further improvements as more sections are gathered to represent the document, during training. To validate this, we measure the document retrieval performance with varying the number of sections per document on the InfoSeek dataset for training. The results shown in Figure 3 then indicate that incorporating more sections raises the MRR@10 score from 7.5 to 15.7. However, this performance boost comes with a clear trade-off; as the number of sections increases, the retriever must process additional end-of-section tokens, leading to higher GPU memory consumption. To balance resource limitations and performance gains, we select four sections per document for all experiments.

**Sections from the same document act as effective negatives to enhance reranker performance.** We explore another method to improve IR effectiveness by leveraging the entire document. Specifically, we investigate the use of sections from the same document as negatives for reranker training, namely In-document. We compare this approach with traditional methods, including Top-K, which selects the top-K retrieved sections as negatives, and In-batch, which uses the positive sections for other samples in the same batch as negatives. After training rerankers with each method, we evaluate section retrieval on the Encyclopedic-VQA dataset. The results shown in Table 6 demonstrate that our In-document approach achieves superior R@1 and MRR@10 scores. This suggests that the use of sections from the same document as negatives provides natural, cost-effective advantages thanks to their high similarity to the positive section. However, it does not consistently outperform the other methods on the R@10 score. We hypothesize that this inconsistency may arise from the strengths of each method: the In-document approach excels at distinguishing sections from the same document, while Top-K and In-batch methods better differentiate sections from different documents.

**BCE loss applied to each section produces the best reranker performance.** In our reranker design, we apply BCE loss using the query concatenated with each document section (Section+BCE). We also explore alternative training objectives to identify the most effective approach for section retrieval in interleaved documents. One such objective is contrastive loss (Contrastive). This approach is similar to the retriever, but the retrieval unit is a section. Additionally, we also explore a variant of the BCE loss (Document+BCE), where, unlike Section+BCE, the query is concatenated with multi-

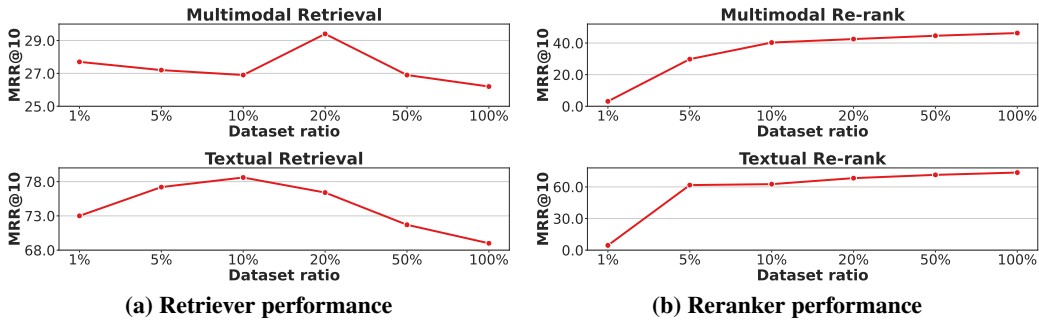

Figure 4: Retrieval performance with different dataset sizes for training. **(a)**: When training a retriever, large datasets rather deteriorate the retrieval performance as it may be overfitted, resulting in low generalization. **(b)**: On the other hand, a larger dataset size is beneficial to training a re-ranker.

ple sections from the same document, including both positive and negative sections. An [EoS] token is appended to each section, and the Document+BCE follows the same BCE loss calculation of the Section+BCE using its [EoS] outputs. This design allows the Document+BCE reranker to leverage the long-context understanding of VLMs to improve section retrieval in interleaved documents.

Then, in Table 7, we compare the section retrieval performance of different reranker designs on the Encyclopedic-VQA dataset. We find that the Contrastive reranker performs the worst, indicating that directly concatenating the query with the section at the input level provides more effective clues for query-section relevance assessment. Notably, this observation is consistent with conventional reranker approaches. Further, the Document+BCE reranker underperforms compared to the Section+BCE reranker, likely due to training constraints. Specifically, while the evaluation phase uses all sections within each document, the training phase is limited to a maximum of four sections per document, with an average of eight sections per document. Such a mismatch may degrade the model's performance. Building on concurrent discoveries (Jiang et al., 2024; Lee et al., 2024), addressing these training constraints will potentially open up new reranker designs that can better handle long, interleaved documents using VLMs, ultimately improving section retrieval performance.

**Rerankers require much larger datasets than retrievers.** We analyze the effect of different dataset sizes for training on retriever and reranker performance. To achieve this, we randomly prune samples in the Encyclopedic-VQA dataset at various ratios and report the performance of models trained on these subsets. In Figure 4 (a), we observe that too many samples can degrade retrieval performance. Also, retrieval of textual queries requires fewer samples to reach its optimal performance compared to multimodal retrieval. Similarly, in Figure 4 (b), section retrieval for multimodal queries requires 10% of the dataset to achieve 80% of the full-dataset performance, while section retrieval for textual queries needs only 5%. These observations suggest that additional modalities increase the need for more data. This accounts for the inferior performance of the interleaved format in the ViQuAE experiments (Table 4 (a)). The ViQuAE dataset, at only 2.2% of the size of Encyclopedic-VQA, may be small for the reranker to effectively learn multimodal query-section alignments. We also observe that section retrieval is more challenging, with more samples improving the reranker's performance. This explains why the ViQuAE reranker has much lower section retrieval scores compared to the one trained on the Encyclopedic-VQA (Table 4 (b)). Given the challenge of obtaining large query-section pair samples, exploring more effective reranker training pipelines is necessary.

## 5 CONCLUSION

In this paper, we introduced IDentIfy, a novel IR framework designed to address the limitations of conventional methods that rely on solely textual content of documents and their segmented passages. Specifically, our approach sits on top of recent VLMs, which enables integration and representation of diverse multimodal content (including text, images, and tables) into a unified document representation. Also, unlike previous strategies that segment documents at the passage level, our method merges these segments to maintain the document's structural coherence, while further introducing a reranking strategy for precise identification of relevant sections. Extensive experiments across various IR datasets demonstrated that IDentIfy consistently outperforms existing baselines, confirming that the interleaved multimodal representation significantly enhances the quality of the document retrieval. We believe IDentIfy represents a crucial step toward more comprehensive and contextually aware IR systems, capable of handling the increasing multimodality of modern information sources.

## REPRODUCIBILITY STATEMENT

Our codes are based on publicly available LLaVA-NeXT (Li et al., 2024). The experimental setup and details can be found in §4 and Appendix A. The experiments are conducted with publicly available datasets (Mensink et al., 2023; Chen et al., 2023; Lerner et al., 2022; Kweon et al., 2023). We have included our codes in the supplementary material and will publicly release our code.

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

# Appendix

**Organization**   The supplementary file is organized as follows: In Appendix A, we explain the implementation details for our experiments. In Appendix B, we outline the limitations of our study.

## A   IMPLEMENTATION DETAILS

Table 8: Information retrieval datasets summary.

| Dataset | Query Modality | Target | Domain | Entities | Section ID | Train | Eval | Test | Corpus size |
|---|---|---|---|---|---|---|---|---|---|
| Encyclopedic-VQA | Text, Text-Image | Text | Species, Landmarks | 17k | o | 177k | 2.2k | 3.8k | 100k |
| InfoSeek | Text-Image | Text | Diverse | 11k | x | 209k | 23k | 74k | 500k |
| ViQuAE | Text, Text-Image | Text | Human | 1k | o | 1.2k | 1.2k | 1.2k | 100k |
| Open-WikiTable | Text | Table | Table | - | o | 3.3k | 0.4k | 0.4k | 1.8k |

**Dataset configuration**   Table 8 summarizes the key properties of the datasets used in our experiment, including query modality, target item, entity domain, number of entities, and whether a section ID is provided to indicate the section containing the answer. Additionally, we provide the number of samples in the training, evaluation, and test splits, as well as the size of the corpus.

**Dataset pre-processing**   In our study, we leverage interleaved multimodal content from Wikipedia documents. However, the existing corpora associated with our IR datasets often lack this content, typically only including the first few words of each document. Therefore, we augment the corpora by downloading the HTML file of each Wikipedia document.

If the dataset provides Wikipedia URLs for its corpus, we use them to download the HTML files. Alternatively, if only entity names are provided, we generate Wikipedia URLs using those names. If a Wikipedia URL is deprecated, we remove the corresponding document from the corpus along with any associated queries. From the HTML files, we extract text, image URLs, and tables. We then split the contents by subtitles in the document where each chunk corresponds to a section. For the images, we use the image URLs to download the corresponding images, removing any invalid URLs. This process produces a dictionary that organizes text, images, and tables by section.

Since downloading the complete contents for all documents across datasets is time- and memory-intensive, we preprocess the subsets of each corpus, including documents relevant to queries in the training, evaluation, and test splits, as well as unrelated entity documents.

## B   LIMITATIONS

Due to the limitations of a single H100 GPU, we represent documents by selecting a limited number of sections and averaging their corresponding embeddings. While this reduces the computational demands, our findings suggest that capturing a broader document context leads to improved retrieval performance. Hence, leveraging the long context window of LVLMs could further enhance document retrieval by capturing more comprehensive information from the full document. Moreover, our reranker design follows the conventional approach of concatenating the input query with individual sections. However, we believe that providing the reranker with all the sections together would allow the model to better leverage the contextual information from the entire interleaved document, potentially resulting in improved performance. In order to fully leverage the interleaved format in the IR system, addressing the issues by reducing the GPU load when processing interleaved documents would greatly boost overall IR performance. We leave these explorations for future work.

