# OpenReview forum: "Unified Multi-Modal Interleaved Document Representation for Information Retrieval"
_ICLR.cc/2025/Conference — ICLR 2025 Conference Withdrawn Submission_

### Official Review · Reviewer_xZc9 · 2024-10-31

**Soundness:** 3
**Presentation:** 2
**Contribution:** 2
**Rating:** 3
**Confidence:** 5

**Summary:**

This paper addresses limitations in document representation for information retrieval (IR) by recognizing that documents can contain multiple modalities—such as text, images, and tables—and that segmenting long documents into discrete passages often hampers the ability to capture overall context and inter-paragraph interactions. The authors propose a novel method that interleaves different modalities in document embeddings, leveraging the capabilities of vision-language models (VLMs) to enhance the representation of multimodal documents. The proposed method aims to improve the effectiveness of document retrieval by better capturing the relationships among various modalities within a single document.

**Strengths:**

1. **Originality**:
   - The paper identifies significant limitations in current document representation methods and proposes an innovative approach to integrate multiple modalities, a relatively underexplored area in information retrieval.

2. **Quality**:
   - The methodology demonstrates a thoughtful integration of VLMs for enhancing document embeddings, showing promise in leveraging advanced models to address multimodal challenges.

3. **Clarity**:
   - The paper is well-structured and articulately presents the limitations of existing approaches, the proposed solution, and the expected impact on information retrieval. This clarity makes it accessible to readers across various backgrounds.

4. **Significance**:
   - By focusing on the multimodal nature of documents, the research has potential implications for various applications in IR, making it a timely contribution to the field as the demand for more sophisticated document processing techniques grows.

**Weaknesses:**

1. **Lack of Novel Contribution**:
   - While the application of VLMs to IR is interesting, the paper lacks substantial novelty beyond their application. Previous works, such as those exploring VLMs in other contexts (e.g., CLIP, BLIP), have already laid the groundwork for similar methodologies.
   - The segmentation of documents into sections does not introduce a new technique; rather, it mirrors existing practices without clear justification for its necessity.

2. **Evaluation and Baselines**:
   - The evaluation framework appears insufficiently rigorous, with limited baseline comparisons provided. The selection criteria for these baselines are not clearly articulated, raising concerns about the validity of the results.
   - There is a notable absence of non-VLM-based evaluations to establish the effectiveness of the proposed method relative to traditional approaches.

3. **Methodological Concerns**:
   - The rationale for dividing documents into sections is not convincingly justified, leaving the impression that it may compromise document representation integrity.
   - The proposed use of representations such as ‘End of Query’ and ‘End of Section’ lacks comparative evidence demonstrating their superiority over alternative representation methods.

4. **Inadequate Discussion of Modality Gap**:
   - The paper does not sufficiently address how the modality gap is resolved, which is critical for understanding the effectiveness of the proposed method.

**Questions:**

1. **Rationale for Sectioning**:
   - Could you clarify the rationale for segmenting documents into sections? What benefits do you envision from this approach that could not be achieved through a holistic document representation?

2. **Alternative Approaches**:
   - Have you considered preprocessing with techniques like CNNs before embedding to retain document-level context without segmenting? How might this impact your findings regarding limitations?

3. **Effectiveness of Representations**:
   - Can you provide empirical evidence or theoretical justification that supports the efficacy of using representations like ‘End of Query’ and ‘End of Section’ compared to other methods?

4. **Baseline Choices**:
   - What criteria did you use to select the baseline models for evaluation? How do these baselines adequately reflect the current state of research in multimodal IR?

5. **Modality Gap Resolution**:
   - How does your approach specifically address the modality gap? Can you elaborate on any mechanisms or metrics used to assess this aspect?

6. **Generalizability of Results**:
   - Since LLaVA-NeXT is highlighted as a strong VLM, how do you anticipate the performance might vary with other VLMs? Have you conducted preliminary analyses to explore this?

---

> ### Author Response · Authors · 2024-11-29
> **Official Comment by Authors**
>
> Dear Reviewer xZc9,
>
> Thank you for your review and helpful comments. We have made every effort to address your concerns.
>
> ---
>
> > #### **[Weakness 1-1] Novelty.**
> > While the application of VLMs to IR is interesting, the paper lacks substantial novelty beyond their application. Previous works, such as those exploring VLMs in other contexts (e.g., CLIP, BLIP), have already laid the groundwork for similar methodologies.
>
> $\rightarrow$ We believe there may be a misunderstanding of the contribution of our work. Our main contribution does not lie in the application of VLMs to IR. Instead, our novel contributions are the proposal of a new task setup of considering multimodal contents interleaved within documents in their most natural format, and the consideration of the contextual information spread in documents (by aggregating section-level representations), with the goal of holistically representing documents for IR. Therefore, the use of VLMs is to operationalize this new idea, and we do not claim novelty on it. Also, the methods that are on top of previous VLMs that you mentioned (such as CLIP or BLIP) are not suitable for our target task of IR, as they can only process a single image or a small chunk of text.
>
> ---
>
> > #### **[Weakness 1-2, 3-1] Justification of segmenting.**
> > The segmentation of documents into sections does not introduce a new technique; rather, it mirrors existing practices without clear justification for its necessity; The rationale for dividing documents into sections is not convincingly justified, leaving the impression that it may compromise document representation integrity.
>
> $\rightarrow$ We would like to clarify that the segmentation of documents into sections is a practical design choice that aligns with conventional methods [1, 2], as it enables the effective handling of long documents. For example, in one of very practical scenarios such as retrieval-augmented generation (RAG), providing segmented sections rather than an entire long article often leads to more accurate query-specific answers, as the model can focus on the most relevant parts of the document without being overwhelmed by extraneous content. In addition to this, segmenting documents allows users or models to access concise, focused information, thereby enhancing the efficiency of processing them.
>
> [1] Mensink et al., Encyclopedic-VQA: Visual questions about detailed properties of fine-grained categories, ICCV 2023
>
> [2] Li et al., Retrieval Augmented Generation or Long-Context LLMs? A Comprehensive Study and Hybrid Approach, EMNLP 2024
>
> ---
>
> > #### **[Weakness 2-1] Baselines.**
> > The evaluation framework appears insufficiently rigorous, with limited baseline comparisons provided. The selection criteria for these baselines are not clearly articulated, raising concerns about the validity of the results.
>
> $\rightarrow$ We clearly outline the selection criteria for our baselines in Lines 306 - 311. Also, they are not limited, which include a range of strategies to represent documents and are indeed far sufficient to validate the advantage of our approach (i.e., demonstrating the importance of considering both contextual and multimodal-interleaved information within documents). Please let us know if you have specific suggestions for additional baselines.
>
> ---
>
> > #### **[Weakness 2-2] Evaluation.**
> > There is a notable absence of non-VLM-based evaluations to establish the effectiveness of the proposed method relative to traditional approaches.
>
> $\rightarrow$ This may be a critical misunderstanding of our work. Our experiment setups and results already include the evaluations that do not consider modalities other than text, and, through this, we already demonstrate the effectiveness of our approach over models with text-only modality. Also, comparing our proposed approach with methods based on other non-VLMs is trivially relevant at best for the task of IR with multimodal documents and queries. This is because not only non-VLM models inherently lack the capability to process multiple modalities, but also differences in base models and their underlying capabilities further make direct comparisons between different approaches unfair and not meaningful.
>
> ---
>
> > #### **[Weakness 3] Representation method.**
> > The proposed use of representations such as ‘End of Query’ and ‘End of Section’ lacks comparative evidence demonstrating their superiority over alternative representation methods.
>
> $\rightarrow$ The use of tokens such as ‘End of Query’ and ‘End of Section’ is a very well-established and standard practice for finalizing embeddings of variable-length queries and passages; therefore, we strongly believe that providing comparative evidence for such a widely accepted approach is unnecessary and not the scope of this work (as our primary focus is not in developing a new tokenization or representation markers for embeddings).
>
> ---

---

> ### Author Response · Authors · 2024-11-29
> **Official Comment by Authors**
>
> > #### **[Weakness 4] Modality gap.**
> > The paper does not sufficiently address how the modality gap is resolved, which is critical for understanding the effectiveness of the proposed method.
>
> $\rightarrow$ This may be a critical misunderstanding of our work as the concept of modality gap is not relevant to our work. Specifically, its concept is commonly used to measure the distance between the representations of two separate modalities in the multimodal representation space [1, 2, 3], whereas, in IR, the multiple modalities within queries and documents are not separately handled. For example, in IR, the query can be either text or a combination of text and image, and the retrieval target (document or section) can be either text or a combination of different modalities, which are not distinctly considered. Therefore, the analysis on the modality gap is clearly unnecessary and inappropriate.
>
> [1] Liang et al., Mind the Gap: Understanding the Modality Gap in Multi-modal Contrastive Representation Learning, NeurIPS 2022
>
> [2] Udandarao et al., Understanding and fixing the modality gap in vision-language models, PhD thesis, University of Cambridge
>
> [3] Shi et al., Understanding the Modality Gap in CLIP, ICLR 2023
>
> ---
>
> > #### **[Question 1] Rationale for sectioning.**
> > Could you clarify the rationale for segmenting documents into sections? What benefits do you envision from this approach that could not be achieved through a holistic document representation?
>
> $\rightarrow$ We thank you for your question. We provide the rationale and benefits for segmenting documents into sections in our response to Weaknesses 1-2 and 3-1.
>
> ---
>
> > #### **[Question 2] Alternative approaches.**
> > Have you considered preprocessing with techniques like CNNs before embedding to retain document-level context without segmenting? How might this impact your findings regarding limitations?
>
> $\rightarrow$ No, we have not considered CNNs as they are designed for image processing, which are not the suitable neural network architectures to process documents. If there are specific techniques or adaptations of CNNs that could effectively handle sequences of tokens comprising text, images, and tables, we would appreciate any suggestions/insights.
>
> ---
>
> > #### **[Question 3] Effectiveness of representations.**
> > Can you provide empirical evidence or theoretical justification that supports the efficacy of using representations like ‘End of Query’ and ‘End of Section’ compared to other methods?
>
> $\rightarrow$ We address this question in our response to Weakness 3.
>
> ---
>
> > #### **[Question 4] Baseline choices.**
> > What criteria did you use to select the baseline models for evaluation? How do these baselines adequately reflect the current state of research in multimodal IR?
>
> $\rightarrow$ Thank you for your question. The baselines considered in our evaluation not only reflect the current state-of-the-art in document (or section) representation for IR but also are carefully selected to ensure a comprehensive validation of the effectiveness of our proposed approach. Specifically, when representing documents for IR, not only recent multimodal IR work utilizes only one image but also most of the conventional IR approaches only consider the textual content within documents, and we include baselines for them in Table 1. From this, we then demonstrate the effectiveness of our approach in representing documents in their interleaved formats with different modalities, for various IR tasks.
>
> ---
>
> > #### **[Question 5] Modality gap.**
> > How does your approach specifically address the modality gap? Can you elaborate on any mechanisms or metrics used to assess this aspect?
>
> $\rightarrow$ We answer this question in our response to Weakness 4.
>
> ---
>
> > #### **[Question 6] Generalizability of Results**
> > Since LLaVA-NeXT is highlighted as a strong VLM, how do you anticipate the performance might vary with other VLMs? Have you conducted preliminary analyses to explore this?
>
> $\rightarrow$ We would like to clarify that the reason we use the same VLM (LLaVA-NeXT) across different approaches is to ensure a fair and consistent comparison. Also, we believe the results with this VLM is sufficient to demonstrate the effectiveness of our approach (incorporating interleaved multimodal information and contextual integration for document representations in IR), and, in achieving this goal, comparing different VLMs and their performances is not the focus of our work. However, we anticipate that using much larger VLMs will enhance the overall performance thanks to their increased capacity.

---

> > ### Comment · Reviewer_xZc9 · 2024-12-02
> > **Reply to authors**
> >
> > Thank you for your reply. I would like to know why you claim that "*the proposal of a new task setup of considering multimodal contents interleaved within documents in their most natural forma*" and "*to represent the document holistically for IR, taking into account contextual information spread throughout the document*" and why this has not been possible before and why you want to make the VLM part of this new task set. Can you explain the novelty and improvements of using these models and methods?

---

> ### Author Response · Authors · 2024-12-02
> **Dear Reviewer xZc9**
>
> We sincerely thank reviewer xZc9 for raising these concerns for a better understanding of the novelty of our work.
>
> ---
>
> > #### **[Concern 1] Novelty Claim**
> > I would like to know why you claim that "the proposal of a new task setup of considering multimodal contents interleaved within documents in their most natural forma" and "to represent the document holistically for IR, taking into account contextual information spread throughout the document"
>
> $\rightarrow$ Our claim is supported by the observation that recent information retrieval (IR) approaches [1, 2, 3] primarily rely on a limited portion of text or a single image to represent a document. However, human-generated documents, such as Wikipedia, naturally contain diverse modalities, including text, images, and tables, providing richer information to represent documents, and no previous works consider these diverse modalities to represent documents. Moreover, the documents are typically long, where each section contributing unique contextual information that enhances the overall document representation.
>
> We would like to emphasize that our work is the first to address this gap by demonstrating the effectiveness of leveraging the interleaved and contextual information within documents over previous IR approaches in diverse scenarios, including document- and section-level IR tasks with uni- and multi-modal queries, suggesting a new task of active utilization of the interleaved multimodal information and contextual information is essential for enhanced IR systems.
>
> [1] Mensink et al., Encyclopedic VQA: VIsual questions about detailed properties of fine-grained categories, ICCV 2023
>
> [2] Caffagni, et al., Wiki-LLaVA: Hierarchical Retrieval-Augmented Generation for Multimodal LLMs, CVPRW 2024
>
> [3] Ma et al., Fine-Tuning LLaMA for Multi-Stage Text Retrieval, arXiv
>
> ---
>
> > #### **[Concern 2] Approach**
> and why this has not been possible before and why you want to make the VLM part of this new task set. Can you explain the novelty and improvements of using these models and methods?
>
> $\rightarrow$ The integration of VLMs into our proposed task setup is enabled by a recent development of VLMs that can process interleaved multimodal content [1, 2]. Hence, we leverage this recent advancement to operationalize the idea of incorporating diverse modalities into a unified representation for improved IR systems.
>
> We would like to clarify our novel contributions, which are 1) a new task for representing documents interleaved with multiple modalities, 2) a general framework that leverages VLMs to operationalize this, and 3) a reranking method to pinpoint the relevant piece within the document.
>
> In our paper, we propose that a simple approach to leverage VLMs to encode interleaved documents into a unified representation can yield superior representation compared to text-only document representation. As shown in Table 1, methods that rely solely on text, such as ‘Entity’ and ‘Text-document’, obtain R@1 scores of 3.1 and 12.5, respectively on the document retrieval task. In contrast, our approach, denoted as ‘+Interleaved’, achieves a significantly higher R@1 score of 20.5. This substantial improvement supports our claim that leveraging both images and tables as well as texts in documents yields effective, holistic document representations, enhancing performance in IR systems.
>
> [1] Li et al, LLaVA-NeXT-Interleave: Tackling Multi-image, Video, and 3D in Large Multimodal Models, arXiv
>
> [2] Zhang et al., InternLM-XComposer-2.5: A Versatile Large Vision Language Model Supporting Long-Contextual Input and Output, arXiv

---

### Official Review · Reviewer_pXLA · 2024-11-03

**Soundness:** 2
**Presentation:** 2
**Contribution:** 3
**Rating:** 5
**Confidence:** 3

**Summary:**

This paper presents Interleaved Document Information Retrieval System (IDentIfy), a document retrieval framework that uses vision-language models (VLMs) to encode the multi-modal document interleaved with textual, visual, and tabular data to perform document retrieval followed by section retrieval. In the document retrieval stage, following the bi-encoder paradigm, the query and document section is separately encoded, and the section embeddings from a document is averaged to form the document embedding. In the section retrieval stage, the authors develop a re-ranker to re-rank sections previously retrieved by the document retriever. Experimental results show that IDentIfy can outperform Entity and Summary baselines as well as textual models.

**Strengths:**

- With the advantages of VLMs, IDentIfy is able to perform effective retrieval on documents interleaved with multiple modalities.
- IDentIfy effectively integrates global information into segmented sections while maintaining efficient training inference.

**Weaknesses:**

- The experiments are conducted on clean, source-avaliable corpus whose documents can be easily segmented into sections according to the subtitles, and then extracted into multi-modal elements. However, real-world data are often presented in compiled files like PDFs. In such scenarios, document division and multi-model data extraction may not be possible. This poses a challenge for IDentIfy in real-world use.
- The presentation of the results in Section 4.3 lacks a main thread, and is difficult to follow. I suggest the authors add an introductory paragraph at the beginning of Section 4.3 and organize the experiments in a clearer structure.
- There are some details in this paper that are not very clear (see Questions).

**Questions:**

- As shown in Table 8, the retrieval target of Encyclopedic-VQA, InfoSeek, ViQuAE is only text. Why does IDentIfy perform better than the Text-document baseline on these datasets?
- The equation on line 240 contains an error: exp is missed in the loss calculation.
- Do “section” and “passage” in this paper mean the same thing? If yes, a sentence could be added stating that the two terms refer to the same thing.
- The terms “document retrieval” and “section retrieval” are confusing. They actually mean the two stages in IDentIfy. But they read like two levels of retrieval granularity, as the experiment presents on line 347.
- How are texts, images, and tables extracted from a section organized into the input to the section encoder? Is it a fixed order of texts, then images, finally tables (as line 210 indicates)?
- What do the authors mean by “combine four images into one”, on line 301?
- How do the authors “consider four sections per document in representing documents” (line 302)? What four, the first four?
- In Table 2 and 1, the passage (section?) retriever performs significantly worse than document retriever (20.5 R@1 for document retriever, 3.9 R@1 for passage retriever, only 19% of the performance of document retriever). Does that mean that the global information plays a so important role, that ignoring it can have a huge impact on retrieval, while a simple embedding averaging can mitigate it effectively? If yes, why can the re-ranker, which doesn’t integrate any global information, offer so much gain (3.9→28.6, closer to 35.1)?

---

> ### Author Response · Authors · 2024-11-29
> **Official Comment by Authors**
>
> Dear Reviewer pXLA,
>
> Thank you for your review and constructive comments. We have made every effort to address your concerns.
>
> ---
>
> > #### **[Weakness 1] Real-world application**
> > The experiments are conducted on clean, source-avaliable corpus whose documents can be easily segmented into sections according to the subtitles, and then extracted into multi-modal elements. However, real-world data are often presented in compiled files like PDFs. In such scenarios, document division and multi-model data extraction may not be possible. This poses a challenge for IDentIfy in real-world use.
>
> $\rightarrow$ Thank you for your constructive feedback. While we acknowledge that documents are sometimes presented in compiled formats (such as PDFs), a more significant portion of publicly accessible documents, particularly those available on the web, are formatted in HTML. Also, we envision that documents represented in PDFs can be converted into HTML from which we can utilize the proposed approach for representing them, and we leave such the direction of handling PDF documents as future work.
>
> ---
>
> > #### **[Weakness 2] Absence of introductory paragraph**
> > The presentation of the results in Section 4.3 lacks a main thread, and is difficult to follow. I suggest the authors add an introductory paragraph at the beginning of Section 4.3 and organize the experiments in a clearer structure.
>
> $\rightarrow$ We appreciate your constructive suggestion. While we tried to provide the main finding of each experiment at the beginning of each paragraph with the bolded sentences, we will improve the presentation and description of Section 4.3 in the revision.
>
> ---
>
> > #### **[Questions 1] Explanation on retrieval targets**
> > As shown in Table 8, the retrieval target of Encyclopedic-VQA, InfoSeek, ViQuAE is only text. Why does IDentIfy perform better than the Text-document baseline on these datasets?
>
> $\rightarrow$ We apologize for the confusion. The retrieval target for those three datasets is the documents interleaved with multiple modalities (such as text, images, and tables); therefore, as our proposed method can holistically consider them over the Text-document baseline that is limited to consider only the text, our method is superior to the baseline.
>
> ---
>
> > #### **[Questions 2] Error on contrastive loss equation**
> >The equation on line 240 contains an error: exp is missed in the loss calculation.
>
> $\rightarrow$ We thank you for pointing it out; we will fix it in the revision.
>
> ---
>
> > #### **[Questions 3] Clear definition of section and passage**
> > Do “section” and “passage” in this paper mean the same thing? If yes, a sentence could be added stating that the two terms refer to the same thing.
>
> $\rightarrow$ Yes, the “section” and “passage” are used interchangeably to refer to the same concept. We will clarify it by adding a footnote (mentioning their equivalence) in our revision.
>
> ---
>
> > #### **[Questions 4] Clear definition of section and passage**
> >  The terms “document retrieval” and “section retrieval” are confusing. They actually mean the two stages in IDentIfy. But they read like two levels of retrieval granularity, as the experiment presents on line 347.
>
> $\rightarrow$ We apologize for the confusion. The section retrieval for the experiment discussed in Line 347 is not the section reranking within the proposed two-stage (retrieval and reranking) pipeline of IDentIfy, but rather it denotes retrieving the sections directly without the document retrieval step. This is the baseline and, in Table 2 (with Line 347), we aim to show that our approach (that first retrieves top-K documents and then identifies relevant sections within them) is superior to the approach of directly retrieving the sections (called section retrieval). We will improve the clarity of it in the next revision.
>
> ---

---

> ### Author Response · Authors · 2024-11-29
> **Official Comment by Authors**
>
> > #### **[Questions 5] Organization of inputs to section encoder**
> > How are texts, images, and tables extracted from a section organized into the input to the section encoder? Is it a fixed order of texts, then images, finally tables (as line 210 indicates)?
>
> $\rightarrow$ Yes, your understanding is correct. As described in Line 210 and illustrated in Figure 2, we organize the different modalities in their conventional order following existing work [1, 2, 3].
>
>
>
> [1] Li et al., LLaVA-Next-Interleave: Tackling Multi-image, Video, and 3D in Large Multimodal Models, arXiv
>
> [2] Zhang et al., InterLM-XComposer: A Vision-Language Large Model for Advanced Text-image Comprehension and Composition, arXiv
>
> [3] Chen et al., ShareGPT4V: Improving Large Multi-modal Models with Better Captions, ECCV 2024
>
> ---
>
> > #### **[Questions 6] Detailed explanation on method**
> > What do the authors mean by “combine four images into one”, on line 301?
>
> $\rightarrow$ As explained in Lines 300 - 302, the “combine four images into one” phrase means we scale each image down to half its original width and height and then combine four scaled-down images into a single composite image (arranged in a grid pattern). This is to achieve our goal of considering several images within a document, since processing each image individually at full resolution significantly increases the token count and imposes an excessive burden on GPU resources, which is not feasible with the best of our resources.
>
>
> ---
>
> > #### **[Questions 7] Detailed explanation on method**
> > How do the authors “consider four sections per document in representing documents” (line 302)? What four, the first four?
>
> $\rightarrow$ We randomly select four sections per document while ensuring the inclusion of the positive section (if it is available in the dataset). This selection of the sections is to balance efficiency and performance during the training of both the retriever and reranker. Specifically, as shown in Figure 3, additional experiments with varying numbers of sections demonstrate that including more sections consistently improves retrieval performance; however, the efficiency trade-off becomes more pronounced as the number of sections increases, highlighting the necessity of this balance in our design.
>
> ---
>
> > #### **[Questions 8] Detailed explanation on method**
> > In Table 2 and 1, the passage (section?) retriever performs significantly worse than document retriever (20.5 R@1 for document retriever, 3.9 R@1 for passage retriever, only 19% of the performance of document retriever). Does that mean that the global information plays a so important role, that ignoring it can have a huge impact on retrieval, while a simple embedding averaging can mitigate it effectively? If yes, why can the re-ranker, which doesn’t integrate any global information, offer so much gain (3.9→28.6, closer to 35.1)?
>
>
> $\rightarrow$ We apologize for the confusion. We would like to clarify the distinctions between the results in Tables 1 and 2, as well as the roles of different retrieval components.
>
> Specifically, Table 1 presents the performance of document retrieval without section-level selection, which aims to showcase the benefits of incorporating interleaved multimodal information. For instance, the comparison between 'Text-document' and '+ Interleaved' demonstrates that integrating multimodal content enhances the retrieval accuracy by providing a more comprehensive representation of the document.
>
> In contrast, Table 2 shows the section retrieval performance. For example, the ‘Passage’ and ‘Document’ settings first retrieve sections and documents, respectively, and then perform reranking over them (to identify the query-relevant section). Also, the 'Passage*' setting (unlike the others) does not employ the reranker over the retrieved sections (i.e., directly selecting sections without reranking). In this regard, we can interpret the results in Table 2 as follows: the improvement from 3.9 R@1 (Passage*) to 28.6 R@1 (Passage with reranking) shows the effectiveness of the reranker in refining section-level relevance; the proposed ‘Document’ approach achieves the best performance thanks to the extra consideration of the global context information during document retrieval before performing section selection.
>
> We will clarify them in the revision.

---

### Official Review · Reviewer_ruy7 · 2024-11-04

**Soundness:** 3
**Presentation:** 3
**Contribution:** 2
**Rating:** 5
**Confidence:** 3

**Summary:**

The paper presents a unified approach to encode document representations for information retrieval, consisting of (1) encoding multi-modal interleaved information in a document; (2) split a document into multiple passages and separately encoding the split passages; then average pooling over the passage embeddings as the document representation. The authors conduct studies on how to fine-tune a VLM retriever and reranker to handle  information retrieval tasks with interleaved document.

**Strengths:**

1. The proposed approach is straightforward. Leveraging the pre-trained VLMs for information retrieval is an important topic.
2. The ablation studies on training a reranker are comprehensive and clearly illustrates the detail on how to train a multimodal reranker.

**Weaknesses:**

1. Although the main claims of the paper (interleaved document embeddings and aggregate representations from sections) are intuitive, the experiments are not fully convinced. (1) Is interleaved document encoding better? No text-only retrievers as baselines are provided. It is reasonable to compare document encoding with and without interleaved images; however, it is also sensible to provide the text-only retriever (such as E5, DRAGON or MistralE5) fine-tuned on the same dataset or zero-shot as the text-only retrieval baseline since using VLM fine-tuned on text-only training data may make the VLM overfitting on the small training data. (2) Is aggregating representation from sections better? The experimental results in Table2 may provide the answer but some settings are not clear to me (See 1. in Questions).
2. Some experimental settings are not clear (See Questions) and I’m somehow a bit confused by the tables in the main experiment. For example, in the same dataset, Encyclopedic-VQA and Enc-VQA, there are document and section retrieval; however, there is no clear explanation of the settings on document and section retrieval (See 3. in Questions).

**Questions:**

1. Clarifying the experimental settings in Table2: If I understand correctly, the comparison of 2nd and 3rd rows is to demonstrate the effectiveness of document retriever (aggregate section embeddings from section retriever) is better than section retriever. However, I cannot find the detailed settings for the 2nd row (i.e., how many documents are passed to rerankers? Since the retrieved unit is section; then, there maybe multiple top-K sections coming from the same document.). For a comparison, my imagination is that the top 25 distinct documents should be first identified from the top-K retrieved sections (where K > 25) before reranking?
2. Why the numbers of the last row from Table1 and Table 2 are different? I assume that they are from the best approach with document retrieval with reranker?
3. For document retrieval, how you conduct reranking? Is the reranking pipeline is still the same as section retrieval? I.e., top-25 documents are provided to the reranker, which reranks all the sections in the top-25 documents and use the maximum score of the section in a document as the score to rerank the document?
4. Have you tried to train a retriever and reranker on all the datasets and check if the ranking models can generalize well across different datasets?

---

> ### Author Response · Authors · 2024-11-29
> **Official Comment by Authors**
>
> Dear Reviewer ruy7,
>
> Thank you for your time and efforts in reviewing our paper, as well as your helpful and constructive comments. We have made every effort to address your concerns.
>
> ---
>
> > #### **[Weakness 1-1] Text-only scenarios.**
> > Although the main claims of the paper (interleaved document embeddings and aggregate representations from sections) are intuitive, the experiments are not fully convinced. (1) Is interleaved document encoding better? No text-only retrievers as baselines are provided. It is reasonable to compare document encoding with and without interleaved images; however, it is also sensible to provide the text-only retriever (such as E5, DRAGON or MistralE5) fine-tuned on the same dataset or zero-shot as the text-only retrieval baseline since using VLM fine-tuned on text-only training data may make the VLM overfitting on the small training data.
>
> $\rightarrow$ Thank you for your thoughtful question. We first would like to clarify that we do have results on text-only retrievers in Table 1, showing that interleaved document encoding is better over them. In addition to this, making comparisons of different retrieval approaches with different base models is trivially relevant at best. This is because different models have different capacities in understanding the documents; therefore, it is challenging to draw the informed conclusion on whether the performance improvement comes from considering extra modalities or from using high-capacity models. Lastly, unlike your concern that training VLMs with text-only training data might result in overfitting, as shown in Figure 4, the VLM is not overfitted to the training data when its size is small.
>
> ---
>
> > #### **[Weakness 1-2] Superiority of aggregating sections.**
> > (2) Is aggregating representation from sections better? The experimental results in Table2 may provide the answer but some settings are not clear to me (See 1. in Questions).
>
> $\rightarrow$ Yes, aggregating representations from sections is indeed better for document representation, which is demonstrated in Table 2 and Figure 3. Specifically, in Table 2, the ‘Documents’ method, which adopts this aggregation strategy, results in superior performance over other models that do not aggregate representations from sections. In addition to this, Figure 3 reinforces this finding by showing a clear trend where increasing the number of sections incorporated into the aggregation improves retrieval performance. In other words, as more sections are considered and their representations are aggregated, the model can more effectively capture the holistic context and interactions across the entire document.
>
> ---
>
> > #### **[Weakness 2] Unclear explanations of experiments.**
> > Some experimental settings are not clear (See Questions) and I’m somehow a bit confused by the tables in the main experiment. For example, in the same dataset, Encyclopedic-VQA and Enc-VQA, there are document and section retrieval; however, there is no clear explanation of the settings on document and section retrieval (See 3. in Questions).
>
> $\rightarrow$ We thank you for pointing them out and apologize for the confusion. We will improve the explanations of experiment settings and results in the next revision. Also, please refer to the more detailed answers for your questions in our subsequent responses.
>
> ---

---

> ### Author Response · Authors · 2024-11-29
> **Official Comment by Authors**
>
> > #### **[Question 1] Unclear settings.**
> > Clarifying the experimental settings in Table 2: If I understand correctly, the comparison of 2nd and 3rd rows is to demonstrate the effectiveness of document retriever (aggregate section embeddings from section retriever) is better than section retriever. However, I cannot find the detailed settings for the 2nd row (i.e., how many documents are passed to rerankers? Since the retrieved unit is section; then, there maybe multiple top-K sections coming from the same document.). For a comparison, my imagination is that the top 25 distinct documents should be first identified from the top-K retrieved sections (where K > 25) before reranking?
>
> $\rightarrow$ We apologize for the confusion. In Table 2, as described in Lines 348-349 and Lines 356-358, the total number of sections considered for reranking is 200 on average for both the ‘Document’ retriever and the ‘Passage’ retriever, where, for the ‘Document’ retriever, we retrieve 25 documents and each document has 8 sections on average (therefore, the total section number is 200 on average). In other words, we first collect 200 sections and then perform reranking, showing that representing the document with aggregated section representations and then identifying the relevant section for the given document is superior to directly performing the section retrieval. We will clarify the description for Table 2 in the next revision.
>
> ---
>
> > #### **[Question 2] Detailed explanation on results**
> > Why the numbers of the last row from Table1 and Table 2 are different? I assume that they are from the best approach with document retrieval with reranker?
>
> $\rightarrow$ This is because Table 1 and Table 2 show results of different retrieval targets. Specifically, Table 1 reports the document retrieval performance (whose goal is to find the top-K relevant documents for a given query); meanwhile, Table 2 reports the section reranking performance (whose goal is to further pinpoint the query-relevant section within the retrieved top-K documents from document retrieval). We will clarify this in the revision.
>
> ---
>
> > #### **[Question 3] Reranking for document retrieval**
> > For document retrieval, how you conduct reranking? Is the reranking pipeline is still the same as section retrieval? I.e., top-25 documents are provided to the reranker, which reranks all the sections in the top-25 documents and use the maximum score of the section in a document as the score to rerank the document?
>
> \$\rightarrow$ Yes, the reranking is the same as the section retrieval. As we explained in our response to Question 2, document retrieval finds the top-K query-relevant documents, while section retrieval scores how relevant the sections from the top-K documents are to the given query. We will make this clearer in the next revision.
>
> ---
>
> > #### **[Question 4] Generalization of reranker**
> > Have you tried to train a retriever and reranker on all the datasets and check if the ranking models can generalize well across different datasets?
>
> $\rightarrow$ In Table 5 (b), we demonstrate the generalizability of the reranker by training it on the Encyclopedic-VQA dataset and testing it on the ViQuAE dataset. Specifically, compared to the performance of the reranker fine-tuned explicitly on the ViQuAE dataset (50.9 in R@10), the reranker without training on it achieves competitive performance (49.0 in R@10), which confirms that the reranker can be generalizable across different datasets.

---

> > ### Comment · Reviewer_ruy7 · 2024-12-02
> >
> > Thanks for your reply. I agree that comparing the proposed model with other text retrieval models is not a fair comparison. But this is still valuable to have an overall understanding that given the current text-image interleaved training data, whether we can outperform existing state-of-the-art text only retrieval models. Even text-of-the-art text retrievers are doing better, I don’t think that the comparison would negatively impact the value of the work. And thank for your clarification, I think the experiment sections should be revised and organized to make the contribution more clearly. For example, I think that merging the overall effectiveness of all the datasets as the same big table and comparing with some variants of the models and other state-of-the-art text or multimodal retrieval models would be more clear and easy to read. Then, the ablation experiments can be shown afterwards to further discuss the impact of each component.

---

### Official Review · Reviewer_boqu · 2024-11-05

**Soundness:** 2
**Presentation:** 2
**Contribution:** 2
**Rating:** 3
**Confidence:** 5

**Summary:**

This paper introduced a novel IR framework, which enables integration and representation of diverse multimodal content including text, images, and tables, into a unified document representation.

**Strengths:**

The motivation of the work is clear and the problem is worth exploring. The proposed methods are technically sound.

**Weaknesses:**

The novelty of the proposed method is limited. The experiment results and discussion sections are not well-presented to demonstrate the effectiveness and benefits of the proposed methods.

**Questions:**

1. The paper proposed to first represent each document as a sequence of sections as $s_i = [V_{S_i}, L_{S_i}, T_{S_i}]$, where $V_{S_i}$,  $L_{S_i}$, and $T_{S_i}$ are visual tokens, text tokens, and table tokens, respectively. Is there a specific reason why concatenate features from different modalities in this way? Have you tried other feature fusion methods?
2. The above mentioned question also exists in section 3.3, is there a specific reason why concatenate query q and s_i? Why not shuffle their positions? Why not choose other feature fusion methods such as inner product, outer product, addition, subtraction, etc.?
3. I was wondering is there a specific reason why choose contrastive loss for training in section 3.2? Have you compared it with conventional cross-entropy loss?
4. In the experiment section, I was wondering have you conducted the experiments with conventional methods such as dual encoder, where the feature embeddings are extracted from LLaVA?
5. In the experimental result and discussion sections, it is worth exploring how much benefits introduced by each modality.
6. Besides, it is worth comparing the additional performance gain introduced by each modality v.s. their extra latency.

---

> ### Author Response · Authors · 2024-11-29
> **Official Comment by Authors**
>
> Dear Reviewer boqu,
>
> Thank you for your review and constructive comments. We have made every effort to faithfully address your concerns.
>
> ---
>
> > #### **[Weakness 1-1] Novelty.**
> > The novelty of the proposed method is limited.
>
> $\rightarrow$ We would like to clarify our novel contributions, which are 1) a new task for representing documents interleaved with multiple modalities, 2) a general framework that leverages VLMs to operationalize this, and 3) a reranking method to pinpoint the relevant piece within the document. Specifically, no previous works consider representing documents interleaved with multimodal content in their natural format, and, to address this gap, we define a new task. Also, to tackle this novel task, we propose a new approach that leverages VLMs to encode interleaved documents into the unified representation.
>
> ---
>
> > #### **[Weakness 1-2] Presentation.**
> > The experiment results and discussion sections are not well-presented to demonstrate the effectiveness and benefits of the proposed methods.
>
> $\rightarrow$ We thank you for raising this concern and apologize for the inconvenience. We believe that the extensive set of experiments that we design is sufficient to validate the effectiveness of our approach, demonstrating that leveraging interleaved and contextual information within the document is effective over previous IR approaches on both document- and section-level IR tasks with uni- and multi-modal queries. We will improve the presentation for our experiment settings and results in the next revision.
>
>
> ---
>
> > #### **[Question 1] Order of modalities.**
> > The paper proposed to first represent each document as a sequence of sections as si=[VSi,LSi,TSi], where VSi, LSi, and TSi are visual tokens, text tokens, and table tokens, respectively. Is there a specific reason why concatenate features from different modalities in this way? Have you tried other feature fusion methods?
>
> $\rightarrow$ Thank you for your question. The conventional VLMs (including the one that we used for our experiments) are pre-trained and fine-tuned with the sequence of images and texts [1, 2, 3]; therefore, we also follow this conventional order to encode documents. We will include the discussion on it in the next revision.
>
>
>
> [1] Li et al., LLaVA-Next-Interleave: Tackling Multi-image, Video, and 3D in Large Multimodal Models, arXiv
>
> [2] Zhang et al., InterLM-XComposer: A Vision-Language Large Model for Advanced Text-image Comprehension and Composition, arXiv
>
> [3] Chen et al., ShareGPT4V: Improving Large Multi-modal Models with Better Captions, ECCV 2024
>
> ---
>
> > #### **[Question 2-1] Order of query and section.**
> > Is there a specific reason why concatenate query q and s_i? Why not shuffle their positions?
>
> $\rightarrow$ The concatenation of query q and section s_i (in the order of q followed by s_i) is a standard way to perform reranking, as shown in multiple previous works [1, 2, 3].
>
>
>
> [1] Ma et al., Fine-Tuning LLaMA for Multi-Stage Text Retrieval, SIGIR ‘24
>
> [2] Beak et al., Direct Fact Retrieval from Knowledge Graphs without Entity Linking, ACL 2023
>
> [3] Gao et al., Rethink Training of BERT Rerankers in Multi-Stage Retrieval Pipeline, ECIR 2021
>
> ---
>
> > #### **[Question 2-2] Fusion methods for reranker.**
> > Why not choose other feature fusion methods such as inner product, outer product, addition, subtraction, etc.?
>
> $\rightarrow$ In Table 7 (a) and (b), we do consider different feature fusion methods for the reranker. This includes ‘Contrastive’, which calculates the cosine similarity between a query embedding and section embeddings, and ‘Document+BCE’, which concatenates sections alongside the query and then performs reranking simultaneously. In contrast, ‘Section+BCE (Ours)’ follows the conventional approach to fuse the information of the query with the section by concatenating their representations, proving that this approach is the best.
>
> ---
>
> > #### **[Question 3] Loss choice for a retriever.**
> > I was wondering is there a specific reason why choose contrastive loss for training in section 3.2? Have you compared it with conventional cross-entropy loss?
>
> $\rightarrow$ We would like to clarify that contrastive learning loss is the standard choice for training retrievers [1], as the primary objective in information retrieval is to distinguish relevant documents from non-relevant ones for a given query. In other words, this IR objective aligns naturally with contrastive learning, which explicitly models the relative distances between documents in relation to the query, unlike cross-entropy loss that treats each document independently and does not inherently optimize for the ranking of documents relative to each other in terms of their relevance to the query.
>
>
>
> [1] Karpukhin et al., Dense Passage Retrieval for Open-Domain Question Answering, EMNLP 2020
>
> ---

---

> ### Author Response · Authors · 2024-11-29
> **Official Comment by Authors**
>
> > #### **[Question 4] Dual encoder structure.**
> > In the experiment section, I was wondering have you conducted the experiments with conventional methods such as dual encoder, where the feature embeddings are extracted from LLaVA?
>
> $\rightarrow$ We apologize for the confusion. The structure of our retriever follows the dual encoder you mentioned. We will make this clear in the final revision of our paper.
>
> ---
>
> > #### **[Question 5] Impact of each modality.**
> > In the experimental result and discussion sections, it is worth exploring how much benefits introduced by each modality.
>
> $\rightarrow$ We would like to note that, in Table 1, we already show the performance gains introduced by each modality. Specifically, in contrast to models only with textual modality, such as ‘Entity’ and ‘Text-document’ methods (that obtain 3.1 and 12.5 in R@1 score), the method with the additional single image (‘+Single-image’) obtains 16.4 in R@1 score, which clearly shows the benefit of including a visual modality in document representations. Also, the ‘+Interleaved’ method (that uses all different modalities including tables in the document) achieves the highest performance of 20.5 R@1 score, supporting our claim that leveraging both images and tables as well as texts in documents yields effective document representations, leading to the performance improvement in information retrieval.
>
>
>
> ---
>
> > #### **[Question 6] Latency.**
> > It is worth comparing the additional performance gain introduced by each modality v.s. their extra latency.
>
> $\rightarrow$ Thank you for your insightful question. We would like to clarify that, in inference time (where we compare the similarity between the given query and documents), no additional latencies are introduced by considering multiple modalities, since each document is encoded into a fixed-size representation regardless of its modality composition.

---

### Note · Authors · 2024-12-14

I have read and agree with the venue's withdrawal policy on behalf of myself and my co-authors.